# Neural Architecture Search by Learning a Hierarchical Search Space

## Abstract

Monte-Carlo Tree Search (MCTS) is a powerful tool for many non-differentiable search related problems such as adversarial games. However, the performance of such approaches highly depends on the order of the nodes that are considered at each branching of the tree. If the first branches cannot distinguish between promising and deceiving configurations for the final task, the efficiency of the search is significantly reduced. In Neural Architecture Search (NAS), as only the final architecture matters, the visiting order of the branching can be optimized to improve learning. In this paper, we study the application of MCTS to NAS for image classification. We analyze several sampling methods and branching alternatives for MCTS and propose to learn the branching by hierarchical clustering of architectures based on their similarity. The similarity is measured by the pairwise distance of output vectors of architectures. Extensive experiments on two challenging benchmarks on CIFAR10 and ImageNet show that MCTS, if provided with a good branching hierarchy, often yielding better solutions more efficiently than other approaches for NAS problems.

## 1 Introduction

Neural Architecture Search (NAS) aims to automate neural architecture design and has shown significant success in the recent years (Zoph & Le, 2016; Liu et al., 2018a; Real et al., 2019), surpassing manually designed Convolutional Neural Networks (CNN) in deep learning (Liu et al., 2018b; 2019; Guo et al., 2020). One-shot NAS methods based on weight sharing (Pham et al., 2018) have reduced computational cost of NAS by avoiding to train individual architectures. A single "supernet", which contains all possible operational/architectural choices, is trained; specific architectures then inherit these shared weights for evaluation. This allows for efficient reuse of training iterations among compatible architectures (Cha et al., 2022; Pham et al., 2018; Bender et al., 2018). However, for vastly different architectures, it may lead to interference, i.e. weights beneficial to one harm others, and vice-versa (Roshtkhari et al., 2023).

Reducing weight sharing during training can alleviate the detrimental effect of this interference. Prior work has explored multiple specialized models for different search space regions (Su et al., 2021b; Zhao et al., 2021a; Roshtkhari et al., 2023) or importance sampling (Liu et al., 2018b; Ye et al., 2022; Xu et al., 2019; Wang et al., 2021a) where the likelihood of an architecture's performance is estimated during training. This estimation is used to identify and sample more frequently promising architectures, gradually reducing possible interference. The challenge of using importance sampling is to robustly estimate and identify superior architectures as early as possible in the training cycle. This early identification is vital for efficient resource allocation by minimizing wasting training resources on unpromising architectures. This requires fast and reliable estimation of the probability distribution in search space with minimal training iteration.

An architecture can be viewed as a graph, where nodes defining the architecture are connected by edges. These nodes present a choice of operations, which are the processes applied to the data (e.g. convolution, fully connected, etc.). For efficient probability estimation in importance sampling, a common assumption is "node independence", where nodes are considered statistically independent variables. For instance, in a neural network, the operation choice for the second layer would not depend on the operation choice in the first layer. This simplifies the architecture probability estimation to a product of nodes' probabilities

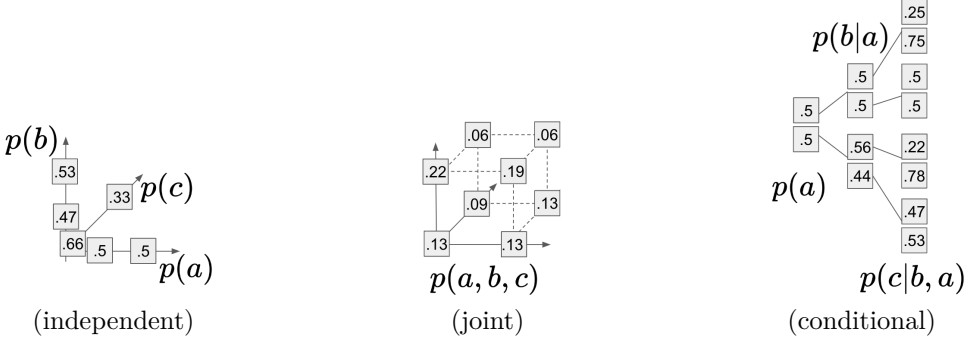

Figure 1: **Probability factorization of 8 architectures.** We show different ways to approximate the discrete probability distribution of architectures for a toy example of search space with N=3 nodes (a,b,c in the figure) each one with O=2 possible operations for a total of $2^3$ architectures. (left) Assuming the nodes independent (as in DARTS (Liu et al., 2018b)) allows the model to estimate only $N \times O$ probabilities. (center) Considering the joint probabilities would require to estimate $O^N$ different probabilities (as in Boltzmann sampling). (right) The joint probability can be factorized into the product of conditional probabilities (in a hierarchy such as in MCTS). This does not reduce the probabilities to estimate, but allows a more efficient exploration of the search space.

(see Fig. 1 (independent)) and reduces the scale of the problem to learning individual node probabilities. Popular differentiable NAS methods (DARTS (Liu et al., 2018b) and followup works (Ye et al., 2022; Xu et al., 2019; Li et al., 2020a)) rely on this assumption. However, overlooking the joint contribution of nodes to performance can lead to a poor node selection for the final architecture (Ma et al., 2023; Zhang et al., 2024b).

Removing node independence assumption requires estimating the joint probabilities of all configurations (see Fig. 1 (joint)). In Single-Path One-Shot (SPOS) methods (Guo et al., 2020; Chu et al., 2021b; Li & Talwalkar, 2020; Stamoulis et al., 2019), at each iteration one architecture from the supernet is sampled, trained, and estimated. While node independence allows updating all node probabilities at each iteration, estimating the joint probability only updates the probability of the sampled architecture. Thus, the full update is inefficient with the estimation cost proportional to the number of architectures, making it unscalable to large search spaces.

To explore more wisely, a compelling option is to factorize the joint probability into conditionals, such as a tree structure for Monte-Carlo Tree Search (MCTS) (Fig. 1 (conditional)) (Świechowski et al., 2023; Costa & Pedreira, 2023). While the number of probabilities to estimate remains the same as the joint probability, it offers several key advantages. A well-designed hierarchical search space enables more efficient tree traversal by reducing the unnecessary exploration of unpromising branches. Prioritizing promising branches can lead to faster convergence, improved solution quality and better scalability. However, standard predefined hierarchies do not guarantee a more efficient exploration, as they are defined by construction and without taking into account the semantic similarity of the corresponding architectures.

While in many MCTS problems the search hierarchy is defined by the sequentiality of the problem (e.g. the moves in chess), for NAS there is no constraint in the order of exploring architectures. Prior work (Zhao et al., 2021b; Wang et al., 2021b; 2020) leveraged this flexibility to reduce the unnecessary exploration by using the classification accuracy of the nodes to partition the search space into "good" and "bad" nodes. However, their approach utilizes MCTS only as a sampler for improving search on a fixed, already learned, recognition model (i.e. the CNN weights), rather than using it during supernet training. In fact, (Zhao et al., 2021b) uses MCTS for searching the best performing model on a supernet pre-trained with uniform sampling, (Wang et al., 2021b) perform MCTS for NAS using the already trained models provided by NAS-Bench-201 (Dong & Yang, 2020), and (Zhao et al., 2024) utilizes both benchmarks and uniformly trained supernet.

In this work, we tackle the more challenging problem of learning the recognition model and the tree partitioning jointly. Utilizing MCTS for sampling during supernet training enables benefits of focused search on favorable

regions. We show that an intelligent factorization of search space is enough to find good architectures, without the need for further regularization (Su et al., 2021a) to compensate for low sampling efficiency.

We propose to factorize the search space in an unsupervised manner based on the semantic similarity of the architectures. The output vector of each architecture, sampled from this supernet, is used to calculate a pair-wise distance matrix of architectures and used for hierarchical clustering and generating tree partitions. The resulting hierarchy implicitly enforces the early nodes of tree to be semantically related, without directly factoring their performance. This approach has the advantage of not relying on architecture performances estimates from supernet which are often inaccurate (Bender et al., 2018; Li et al., 2020c; Zhang et al., 2020) or the additional cost of iteratively refining the tree to account for this inaccuracy (Roshtkhari et al., 2025b;a). This performance agnostic clustering enhances learning and consequently accelerates the search process.

The main contributions of our work are the following:

- We present a new understanding of classical choices of models and strategies for NAS based on the sampling approach and the estimation of the underlying probability of a given architecture. We show that overly restrictive assumptions (e.g. node independence) enables faster estimation, but converges to suboptimal solutions. In contrast, adopting more realistic approach based on conditional probabilities and a hierarchical search space can lead to better solutions.

- We propose an efficient method to sample from the search tree by learning to build a good hierarchy that avoids low-performing architectures. To build this hierarchy, we evaluate several approaches and show that the most promising is based on pairwise distances between architectures, derived from a supernet after MCTS warm-up phase with uniform sampling.

- We empirically validate our findings on two NAS benchmarks on CIFAR10 dataset and MobileNet ImageNet search space. Our results show that the proposed approach improves over previous MCTS approaches and can discover promising architectures within a limited computational budget.

## 2 Related Work

**One-shot methods**  One-shot methods (Pham et al., 2018; Bender et al., 2018) have become very popular in NAS (Liu et al., 2018b; Guo et al., 2020; Su et al., 2021a) due to their efficiency and flexibility. Generally, the training of the supernet and searching for the best architecture can be decoupled (Guo et al., 2020; Wang et al., 2021b) or performed simultaneously (Liu et al., 2018b). In the former, the search can be performed by various methods, such as random search (Bender et al., 2018; Li & Talwalkar, 2020), evolutionary algorithms (Guo et al., 2020) or MCTS (Wang et al., 2021b) and the supernet is static during this phase. The latter alternates between training the supernet and updating the reward to guide the search, such as updating architecture weights in differentiable methods (Liu et al., 2018b), controller in RL (Pham et al., 2018) or probability distribution in MCTS (Su et al., 2021a). However, the quality of supernet as a proxy for architecture evaluation has been the subject of scrutiny in recent years, with various results in different settings (Yu et al., 2019; Wang et al., 2021c; Zela et al., 2019; Termritthikun et al., 2021; Zhang et al., 2024a). A proposed solution is to explicitly reduce the weight sharing among architectures by non-hierarchical factorization of the search space (Roshtkhari et al., 2023; Su et al., 2021b; Zhao et al., 2021a; Ly-Manson et al., 2024; Li et al., 2025). Nevertheless, in general, these methods are computationally more expensive as they require training additional models. Tree-based approaches can be viewed as a form of hierarchical factorization of the search space that reduces weight sharing.

**Node independence**  Early NAS methods using reinforcement learning (Zoph & Le, 2016; Pham et al., 2018), or evolution (Real et al., 2019; Sun et al., 2020; 2019) do not treat nodes as independent, but they were computationally expensive. More efficient and widely adopted differentiable methods (based on DARTS (Liu et al., 2018b)) use back-propagation to learn both node weights (probabilities) and supernet weights. However, one of their known issues is that the learned weights for the nodes fail to accurately reflect their contribution to the ground truth performance and ranking (Wang et al., 2021c; Yu et al., 2019). While several studies have attempted to improve DARTS (Xu et al., 2019; Ye et al., 2022; Chu et al., 2020b; Chen

et al., 2021c; Chu et al., 2020a), few have directly explored the contribution of node independence assumption toward this problem (Ma et al., 2023; Xiao et al., 2022).

Shapley-NAS (Xiao et al., 2022) highlights the underlying relationship between nodes by showing that the joint contribution of node pairs often differs from the accumulation of their separate contributions, due to potential collaboration/competition. They propose to reweigh the learned architecture weights using Shapley value. However, estimating the Shapely value can be costly as it requires training the supernet multiple times. ITNAS (Ma et al., 2023) explicitly models the relationship between nodes via a transition matrix and an attention vector that denotes the node probability translation to successor nodes. The matrix and vector are optimized in a bi-level framework alongside node probabilities. However, this method is currently limited to cell-level and extension to a more general macro search space is not straightforward (for details about macro and cell-based search spaces see appendix A).While these works try to incorporate node dependencies within differentiable NAS framework, an alternative approach is to directly learn either joint or conditional probabilities of the sampled architectures.

**Monte-Carlo tree search**  MCTS with Upper Confidence bound applied to Trees (UCT) (Auer et al., 2002) has been used previously for NAS (Negrinho & Gordon, 2017; Wistuba, 2017). AlphaX (Wang et al., 2019) used a surrogate network to predict the performance of sample architectures, and MCTS to guide the search. TNAS (Qian et al., 2022) aims to improve the exploration of the search space by using a bi-level tree search that traverses layers and operations iteratively. However, the binary tree is manually designed and unbalanced (Le et al., 2024). LaMOO (Zhao et al., 2021b) and LaNAS (Wang et al., 2021b) aim to tackle the problem of finding the best architecture and assume that the deep learning model is given either from a trained supernet (Wang et al., 2021b) or using precomputed benchmarks (Zhao et al., 2021b). In our work, we instead aim at training the deep learning model and finding the corresponding optimal architecture with MCTS in the same optimization, which makes the problem more challenging.

The closest works that jointly performs the model training and architecture search with MCTS are Su et al. (2021a) and Roshtkhari et al. (2025b). Su et al. (2021a) proposes to construct a tree branched along operations. During the training, hierarchical sampling is used for node selection, updating the supernet weights and the reward. Node statistics are then used to update a relaxed UCT probability distribution. However, the tree design is manual, and a regularization method (named "node communication") is required to compensate for insufficient visits of nodes. Iterative-MCTS (Roshtkhari et al., 2025b) constructs the tree based on accuracy estimates from the supernet and iteratively refine the structure throughout training.

**Generalizable and training-free NAS**  An emerging approach is to eliminate task-specific search overhead by developing NAS methods that generalize across tasks without retraining. Recent work has re-framed NAS as a generative rather than a search task. Agiollo & Omicini (2022) propose a meta-learning framework that uses an adversarial graph neural network to directly generate high-performing neural architectures by learning the underlying structural distribution of successful networks, effectively bypassing the need for iterative search. Lee et al. (2021) propose to generate graphs (or architectures) from a given dataset via a cross-modal latent space. These methods offer rapid cross-task transfer, they often rely on pre-training on meta-datasets (such as existing benchmarks like NAS-Bench-201 (Dong & Yang, 2020)) to learn the mapping between data distributions and network performance. Complementary, training-free methods utilize zero-cost proxies (Mellor et al., 2021; Abdelfattah et al., 2021), such as gradient-based metrics (Li et al., 2023) or saliency scores (Lee et al., 2018) to rank architectures at initialization, bypassing the expensive supernet training phase entirely. However, they often show inconsistent rank correlation and poor prediction capabilities (White et al., 2021b; Yang et al., 2023).

## 3  Training by Sampling Architectures

We use single-path supernet training, in which, given a neural model $f$ (e.g. a CNN) for each mini-batch of training data $\mathcal{X}$ and corresponding annotations $\mathcal{Y}$, a different architecture $a$ from the search space $\mathcal{S}$ is

sampled and back-propagated with the following loss:

$$\mathcal{L}(f_a(\mathcal{X}, w), \mathcal{Y}) = \sum_{(x,y) \in (\mathcal{X}, \mathcal{Y})} l(f_a(x, w), y), \tag{1}$$

where $l$ is the sample loss (for instance cross-entropy) and $w$ are the network weights. Training speed and model performance can vary significantly depending on how $a$ is sampled. To prevent overfitting on training data in importance sampling methods, we use validation accuracy as the reward to estimate the probability distribution; and use on-line estimation on mini-batches to accelerate the process. In the following, we present some of the most common sampling techniques, from uniform sampling to our proposed approach.

**Uniform sampling** In the simplest and the original approach of SPOS (Guo et al., 2020), an architecture is sampled uniformly: $a \sim \mathcal{U}(|\mathcal{S}|)$, where $|\mathcal{S}|$ denotes the cardinality of the search space. Despite its simplicity, this sampling method is unbiased and given enough training, all architectures will have the same importance. This method also requires no additional information storage during the training, and in principle can accommodate even very large $|\mathcal{S}|$. In practice, however, the equal importance can present two possible challenges: i) With strong weight sharing (i.e. most weights are shared among many configurations), the same weight adapts to very different architectures, leading to destructive interference and therefore low performance (see (Zhang et al., 2020; Roshtkhari et al., 2023)). ii) If weight sharing is minimal, architectures are almost independent and the training time would increase proportionally to $|\mathcal{S}|$. While uniform sampling may be combined with search space partitioning in a trade-off (Roshtkhari et al., 2023; Su et al., 2021b; Zhao et al., 2021a), it requires training multiple models, demanding higher memory consumption and computational cost. A different direction is to find ways to prioritize the sampling of the more promising architectures.

**Importance sampling with independent probabilities** A simplest way to estimate the importance of each operation is assuming each node $a_i$ as independent. Thus, the probability of an architecture $a$ with $t$ operations is approximated as $p(a) = p(a_1)p(a_2)...p(a_t)$. This simplifying assumption improves sampling efficiency by reducing the number of probabilities to estimate. However, the solution quality is compromised, as it disregards the joint influence of nodes on the performance (Ma et al., 2023; Zhang et al., 2024b).

**Importance sampling with joint probabilities: Boltzmann sampling** In Boltzmann sampling, architecture $a$ is sampled from a Boltzmann distribution with probability $p(a) \propto \exp(\frac{\epsilon_a}{T})$, where $\epsilon_a$ is the estimated rewards (here accuracy) of $a$, and $T$ is the temperature. Sampling is performed with an annealing temperature, starting from a high value (almost uniform), so that the initial phase of the training is unbiased, to a low value (almost categorical) so that the training focuses on high-performing architectures. While more efficient than uniform sampling, estimating $\epsilon_a$ remains time consuming, particularly for large search spaces, and it is difficult to balance exploration/exploitation trade-off in Boltzmann exploration (Cesa-Bianchi et al., 2017).

**Sampling with conditional probabilities: Tree search** Instead of a flat vector of probabilities, we consider a tree of conditional probabilities: $p(a) = p(a_t|a_{(t' \leq t-1)})p(a_{t-1}|a_{(t' \leq t-2)})...p(a_1)$. Each $a_t$ represents a level of the tree that partitions the set of possible architectures into disjoint subsets. The commonly used structure of the tree (Fig.2 (b)) is defined by factorizing the model architectures layer by layer (Su et al., 2021a), starting from the first layer to the last one. Assuming for simplicity a symmetric binary tree, the first level would split the configurations into two disjoint groups. This process of recursive partitioning continues at each subsequent level. With uniform sampling, at each iteration the nodes in level $t$ are sampled with probability $(1/2)^t$.

Thus, probability estimates for early nodes tend to be sufficiently accurate because of high sampling rate. In contrast, for Boltzmann, the sampling rate is $1/|\mathcal{S}|$, which can be extremely small for a large search space. However, if initial nodes maintain a near-uniform probability distribution (not sufficiently discriminative), the estimation of the posterior nodes would suffer from low sampling rate. A possible solution is the regularization proposed by Su et al. (2021a), in which at each update of a specific node, all other equivalent nodes (nodes at the same level with the same operation) are updated similarly using an exponential moving average.

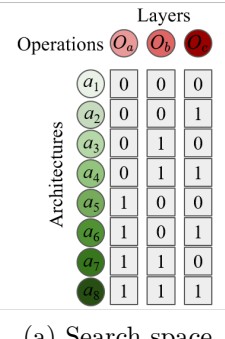
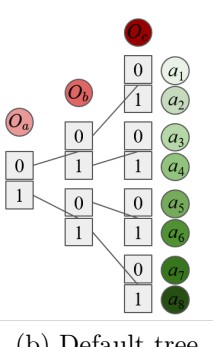
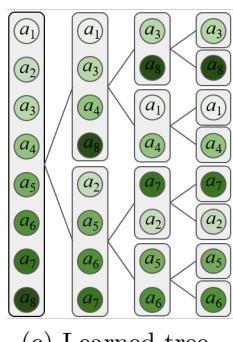


(a) Search space        (b) Default tree        (c) Learned tree


Figure 2: **Comparison of the standard tree structure and our learned structure on a 3 binary operations search space.** (a) The search space consists of architectures with 3 binary operations $(o_a, o_b, o_c)$ which leads to 8 architectures $(a_1, a_2, ..., a_8)$. (b) The default tree structure uses the order of operations (e.g. layers) to build the tree, however this is not necessarily optimal. (c) Our learned tree structure uses a tree that is generated by an agglomerative clustering on the model outputs.

This mechanism of multiple simultaneous updates allows for faster probability estimation and more efficient exploration.

While seemingly an adequate solution, this regularization comes with limitations: i) It assumes homogeneous tree structure at each level, i.e. all nodes at a given level have similar structure (identical children), which limits the approach to specific search spaces. For instance, this approach would not be suitable for search spaces where the operations in a node are conditioned to the choice of operation at the previous node. ii) Reusing the same probabilities for equivalent nodes implies treating nodes independently. In this case, the node independence assumption is enforced in a soft way by a regularization coefficient. Thus, the method attempts to find a compromise between full independence and conditional dependence, but it remains unclear if this trade-off is optimal.

## 4 Our Approach

Our approach tackles the low sampling rate issue in posterior nodes of MCTS from another perspective. We aim to find node ordering that enables early decisions (early nodes) to effectively separate large groups of potential solutions from bad ones. This requires imbalanced probabilities for early nodes that concentrates training on a reduced set of architectures. This increased sampling rate improves local performance estimation from supernet within this region, as interference from lower-performing architectures are reduced. Considering the example in Fig. 1 (conditional), instead of building the tree starting from node $a$, one could start from node $c$. The imbalanced probability of $c$ allows for more efficient sampling compared to $a$. This reordering is possible because, unlike problems in which the ordering of nodes is defined as part of the problem, NAS only cares about the final architecture and not the specific order used to reach it. Therefore, rather than employing a predefined hierarchy, we propose to learn an improved node ordering for MCTS.

In this section, we present a different approach to build a tree of architectures. As shown in Fig. 2 (c), our tree is built based on hierarchical clustering of architectures. Each node of the tree represents a cluster of architectures, going from the root that contains all architectures in a single cluster to the leaves that each contains a single architecture. Through this approach, we release the tree from dependence on the binary operations and allow any possible hierarchical grouping of architectures.

**Hierarchy.** The root node of the tree correspond to the entire search space $\mathcal{S}$. At each node $j$, we partition the search space into two disjoint subsets $\mathcal{S}_j = \cup_{k \in (1,2)} \mathcal{S}_k$. We want to recursively split those into subsets, until reaching the leaf nodes that contain only one architecture. For the sake of simplicity we limited ourselves to binary splits, so that we obtain a binary tree. This splitting can be performed with different heuristics. The simplest using validation performance (Roshtkhari et al., 2025b), with $\mathcal{S}_j = \cup_{k \in (good,bad)} \mathcal{S}_k$, such that for accuracy: $Acc(f_{a \in \mathcal{S}_{good}}(x, w)) > Acc(f_{a \in \mathcal{S}_{bad}}(x, w))$ for each node for validation samples $x \in \mathcal{X}_{val}$.

While this approach seems meaningful, it relies on the assumption that supernet can accurately differentiate $Acc(f_a(x, w))$ for various architectures. However, the supernet obtained from uniform sampling often does not adequately reflect the true ranking of architectures. Instead, in this work we propose to use an agglomerative clustering based on model distances as detailed in the next paragraph.

**Clustering.** A common method to identify hierarchical relationships of data is using agglomerative hierarchical clustering (Murtagh & Legendre, 2014). Given a set of leaf nodes (architectures $a \in \mathcal{S}$) and distances $d(a_i, a_j)$, the algorithm iteratively merges pairs of clusters $C_{new} = C_j \cup C_j$ that are the closest based on a linkage criteria $(C_i, C_j) = \arg \min L(C_k, C_{l \neq k})$. The sequence of these merges defines the hierarchical structure for tree. To measure distance, we seek a representation of architectures that adequately summarizes and captures the relevant information for the given task, independent of the quality of supernet. The output vector of the supernet $f_a(x, w)$ for a given set of validation samples $x \in \mathcal{X}_{val}$ is a suitable representation, as distances between architectures would have semantic (functional) meaning (Klabunde et al., 2025) in the class space (See sec. 5.2 for ablation studies on tree design). With this learned representation, a clustering algorithm can then be used to build the hierarchy based on the distances between architectures.

To ensure that output vectors provide a meaningful basis for architectural similarity (or distance), a partial training of supernet is necessary to allow for adequately capturing how different architectures process data. In our methods, we use a warm-up phase for MCTS using uniform sampling. At the end of warm-up phase, we use the uniformly trained supernet to obtain output vectors by performing a forward pass with a mini-batch of validation data and concatenating the outputs. Next, the pairwise distances of architectures are calculated and the resulting distance matrix is used for hierarchical agglomerative clustering to construct a binary tree. We argue that this method allows us to effectively cluster architectures with similar overall functionality, even if they might differ in their structure in term of their operations. For more details about the construction of the tree, see algorithm 1.

**Search and training.** We use a modified MCTS for both supernet training and search. Similar to Su et al. (2021a) and Wang et al. (2021b), the tree is fully pre-expanded and thus expansion and roll-out stages of traditional MCTS are skipped. Similar to Su et al. (2021a), we use Boltzmann sampling for the selection stage. The Boltzmann distribution allows sampling proportional to the probability of the reward function, producing better exploration (Painter et al., 2024), which is fundamental for good training of the model. For a node in the tree $a_i$, we perform importance sampling with a Boltzmann sampling relative to the node:

$$p(a_i) = \frac{\exp(R(a_i)/T)}{\sum_j \exp(R(a_j)/T)}, \tag{2}$$

where $R$ is our reward function and $T$ is temperature, determining the sharpness of distribution and the normalization sum on $j$ is taken on the sibling nodes of $i$. Training consists of sampling each level of the tree from the root to the leaf, followed by a gradient update of the recognition model $w$ with the sampled architecture $a$ on a mini-batch of training data $\mathcal{X}_{tr}$ and an update of the reward function for the explored nodes, from the leaf to the root of the tree based on the obtained accuracy of the architecture on a mini-batch of the validation set $\mathcal{X}_{val}$. To balance exploration/exploitation, the Upper Confidence bound applied to Trees (UCT) (Kocsis & Szepesvári, 2006) is used as reward for sampling. Considering a node in tree $a_i$, our reward is defined as:

$$R(a_i) = C(a_i) + \lambda \sqrt{log(|parent(a_i)|)/|a_i|}, \tag{3}$$

in which the second term is for exploration. We use function $|a_i|$ to show number of times node $a_i$ is visited, with the constant $\lambda$ controlling the exploration/exploitation trade-off. We define $C$ as:

$$C(a_i) \leftarrow \beta \, C(a_i) + (1 - \beta) \, Acc(f_a(\mathcal{X}_{val}, w)), \tag{4}$$

which is a smoothed version of the validation accuracy of architecture $a$, with smoothing factor $\beta$, to account for the noisy on-line estimation on mini-batches.

---

**Algorithm 1:** Simplified pseudo-code of our training pipeline.

---

**Input** : $\mathcal{S}$: Search Space; $\mathcal{X}_t, \mathcal{X}_v$: mini-batches of training and validation data; $f_a$: model with architecture $a$; $w_p, w$: weights of the model after warm-up and and final model initialized randomly; $e_{pt}, e_{MCTS}$: warm-up and MCTS iterations; $\alpha$: learning rate; $\beta$: smoothing factor; $\lambda$ : exploration parameter of UCT.

*#MCTS warm-up*
**while** *epochs* $\leq e_{pt}$ **do**
    $a \leftarrow$ sample from $\mathcal{U}(|\mathcal{S}|)$
    $w_p \leftarrow w_p - \alpha \nabla_{w_p} \mathcal{L}(f_a(\mathcal{X}_t, w_p))$
*#build the search tree*
**for** $a^i \in \mathcal{S}$ **do**
    Output vector $o^i \leftarrow f_{a^i}(\mathcal{X}_v, w_p)$
Distance matrix $D_{ij} = dist(o^i, o^j)$
Binary tree $\mathcal{T} \leftarrow aggl\_clustering(D)$
*#main training with MCTS*
**while** *epochs* $\leq e_{MCTS}$ **do**
    *#sample an architecture a*
    $a = []$
    $node = $ "root"
    **push**($a$,*node*)
    **while** $not(is\_leaf(node))$ **do**
        $node \leftarrow sample(next(node))$ with Boltzmann sampling as in Eq.(2)
        **push**($a$,*node*)
    *#update model w and accuracy*
    $w \leftarrow w - \alpha \nabla_w \mathcal{L}(f_a(\mathcal{X}_t, w))$
    $accuracy \leftarrow Acc(f_a(\mathcal{X}_{val}, w))$
    $node \leftarrow$ **pop**($a$)
    *#update rewards*
    **while** $not(is\_root(node))$ **do**
        $parent \leftarrow$ **pop**($a$)
        $C(node) \leftarrow \beta\ C(node) + (1 - \beta)\ accuracy$
        $R(node) \leftarrow C(node) + \lambda \sqrt{log(|parent|)/|node|}$
        $node \leftarrow parent$

**Output** : Best architecture from $\mathcal{T}$ by sampling with $\lambda = 0$

---

**Training Algorithm** The training pipeline for our method is shown in algorithm 1. First, a warm-up with random sampling of the architectures is performed in order to train an initial model $f$ with parameters $w_p$. With this model we build a pairwise matrix $D_{i,j}$ that measures the distance of configuration $i$ and $j$ on the output space of the model. With this matrix, we use agglomerative clustering to build a binary tree that represents the hierarchy that will be used for the subsequent MCTS training. During training, an architecture is sampled from the tree, where at each node Boltzmann sampling with the learned probabilities is used. Then, this architecture is used to update the model on a mini-batch of training data (for simplicity we did not include momentum in the gradient updates) and to estimate its accuracy on a validation mini-batch. The validation accuracy is smoothed with an exponential moving average and used as reward with UCT regularization for updating the node probabilities of the sampled architecture. To search for the final architecture after training, we sample $k$ architectures without exploration ($\lambda = 0$) and rank them based on their performance on validation dataset, selecting the best as the final architecture.

## 5 Experiments

We evaluate our method on two macro search space benchmarks using CIFAR10 dataset (Krizhevsky et al., 2009), and on ImageNet (Russakovsky et al., 2015) with MobileNetV2-like (Sandler et al., 2018) search space.

Table 1: **Accuracy and ranking on the Pooling benchmark on CIFAR10.** We report the found architecture (represented with number of layers per feature map sizes), best and average of 3 training accuracy and ranking and search time for different methods.

| Method | Arch. | Best Acc. | Avg. Acc. | Best Rank | Avg. Rank | GPU (hour) |
|---|---|---|---|---|---|---|
| Default Arch. | [4,3,3] | 90.52 | - | 15 | - | - |
| Uniform | [4,3,3] | 90.52 | $90.40 \pm 0.08$ | 15 | 17 | 1.5 |
| MCTS | [4,4,2] | 90.85 | $90.57 \pm 0.21$ | 12 | 15.3 | 2 |
| Boltzmann | [3,5,2] | 90.88 | $90.51 \pm 0.12$ | 11 | 15.3 | 3 |
| Independent | [3,5,2] | 90.88 | $90.86 \pm 0.01$ | 11 | 11.7 | 2 |
| Mixtures (Roshtkhari et al., 2023) | [5,3,2] | 91.55 | $91.36 \pm 0.27$ | 4 | 5 | 6 |
| MCTS + Reg.(Su et al., 2021a) | [6,1,3] | 91.78 | $91.42 \pm 0.11$ | 3 | 3.6 | 2 |
| MCTS + Learned (ours) | [6,2,2] | 91.83 | $91.72 \pm 0.12$ | 2 | 3 | 2 |
| Best | [7,1,2] | 92.01 | - | 1 | - | - |

We compare our proposed method with various sampling methods discussed in sec. 3. For MCTS methods, we start by a warm-up phase of uniform sampling. After this phase, we utilize learned representation of architecture for hierarchical clustering. We also use the recorded statistics to calculate UCT (eq. 3) and sample using eq. 2. For MCTS default tree, used by Su et al. (2021a), each layer of CNN is considered as a level of tree, with operations providing the branching. We compare with this method with and without soft node independence assumption (regularization). For further experimental details see appendix B.

## 5.1 Pooling Search Space

To thoroughly investigate our proposed method, we use Pooling search space (Roshtkhari et al., 2023; Javan et al., 2023), a small yet challenging CIFAR10 benchmark consisting of 36 Resnet20-like (He et al., 2016) architectures. The only architecture parameter optimized is feature map sizes at each layer determined by placement of downsamplings operations. Specifically, each layer has the binary choice of performing downsampling (pooling) or maintaining resolution (identity). The main challenge of this search space lies in full weight sharing across all architectures that contributes to the inadequacy of several common methods (Roshtkhari et al., 2023).

To demonstration each architecture, we use number of layers per feature map sizes (e.g. [4,3,3] meaning 4/3/3 layers in high/middle/low resolution). Our method achieves better results in comparison within similar or shorter search time (Table 1). While for MCTS (default tree design), the regularization proposed in Su et al. (2021a) appears to help, our method obtains better performance without requiring regularization.

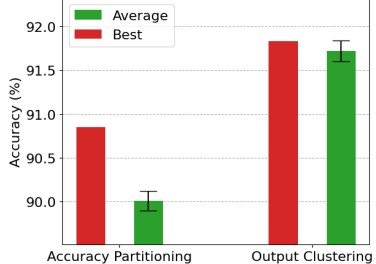

## 5.2 Ablation Studies

We conduct several ablations to assess the convergence and performance of our method and explore alternative ways to design the hierarchy in Pooling search space.

Figure 3: **Best and average accuracy for partitioning** based on validation accuracy vs. output clustering.

**Partitioning with accuracy.** We compare our method with using architecture accuracy to partition search space (See sec. 4 (Hierarchy)). We used validation accuracy estimated from the supernet to construct a binary tree by recursively splitting architectures based on their accuracy (top 50% vs. bottom 50%). The resulting MCTS on this tree is notably worse (Figure 3). This highlights the shortcoming of using inaccurate supernet performance estimation as the basis for search space factorization.

**Branching quality and NAS convergence.** To demonstrate the crucial role of learning tree structure for MCTS efficiency and convergence, we compared MCTS using our learned tree with default tree, and with a binary tree created from a random distance matrix (see Fig. 4). For fair comparison we use the number of iterations for all methods. While the average accuracy of the supernet increases in general over epochs, the

branching quality can affect how the search space is explored. After a warm-up period for UCT with uniform sampling (epoch 340), our tree converges more rapidly to higher accuracy compared to default and random tree. This highlights the importance of a meaningful hierarchy for efficient exploration and faster convergence in NAS.

**Alternative architecture representations.** We explore two types of encodings as zero-cost representation alternatives to supernet outputs. While distances based on outputs measure semantic (functional similarity) distances, using encoding measures structural distances. Representing an architecture as a graph, the general encoding is the adjacency matrix, corresponding to the edges (or one-hot encoding of operations per layer). We also consider the categorical representation of the one-hot encoding as vectors as other alternative.

Table 2 shows diminishing results compared to using outputs in both cases. We note that naively calculating the distances (same weight for all layers) is equivalent to considering each layer as an independent variable, while in fact the posterior layers have less importance than early layers. Therefore, we also consider an exponentially weighted encoding which leads to slight performance increase.

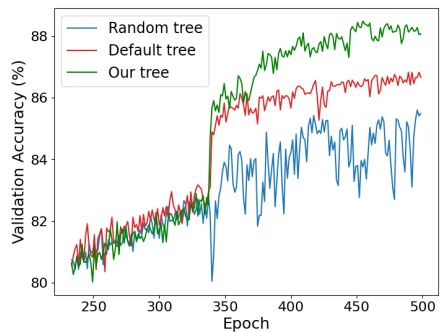

Figure 4: **Accuracy over epochs for several training strategies.** After warm-up, our approach is constantly better than default or a random tree.

### 5.3 NAS-Bench-Macro Search Space

This benchmark consists of 8 layers of MobileNetV2 (Sandler et al., 2018) blocks with operation choices {*Identity, MB3_K3, MB6_K5*} resulting in $3^8 = 6,561$ architectures. Table 3 presents results of several sampling based NAS methods. Our approach finds the best architecture in this search space. Note that for each method, we use the best reward and compare different rewards in appendix C.1.

### 5.4 Search on ImageNet

ImageNet (Russakovsky et al., 2015) consists of 1.28 million training images in 1000 categories. In our ex-

Table 2: **Comparing zero cost branching methods.** We consider one-hot encoding of operations per layer or categorical vector representation. We also consider an exponential weighting scheme to increase the influence of earlier layers on distance.

| Encoding | Weighted | Best Arch. | Best Acc. | Avg. Acc. |
|---|---|---|---|---|
| Vector | | [2,5,3] | 90.89 | $90.63 \pm 0.68$ |
| Vector | ✓ | [3,4,3] | 90.92 | $90.81 \pm 0.15$ |
| One-hot | | [5,2,3] | 90.96 | $90.60 \pm 0.22$ |
| One-hot | ✓ | [5,1,4] | 91.05 | $90.78 \pm 0.21$ |

periments, we use 50k images of validation set as the test data to compare with other methods. To accelerate our training we use mixed precision and FFCV (Leclerc et al., 2023) library. We use similar macro search space to previous works (Su et al., 2021a; You et al., 2020; Chu et al., 2021b; Guo et al., 2020), based on MobileNetV2 blocks with optional Squeeze-Excitation (SE) (Hu et al., 2018) module. The total operation choice per layer is 13 resulting in $13^{21}$ search space size for 21 layers. The choices are convolution kernel size of $\{3, 5, 7\}$ and expansion ratio of $\{3, 6\}$, identity and SE option.

Similar to Su et al. (2021a), we use FLOPS reduction by defining a budget for our search. Leveraging the fact that FLOPs can be used as a zero-cost proxy for architecture performance (Chen et al., 2021b), we sample only within a certain range of target budget ($[0.99, 1] \times$ budget). In Table 4, we compare our method with several NAS approaches with two FLOPs budget, on the upper and middle section of the table. In the bottom section, we provide several methods with final architectures outside our target FLOPs budgets. The methods without references are our implementations and baselines. Within FLOPs budget, our method shows improved performance often with lower computational cost than compared methods. We attribute this advantage to the learned structure of the tree, which allows quicker learning of the promising architectures.

Table 3: **Accuracy and ranking on NAS-Bench-Macro.** We compare our method and several approaches in terms of best, average accuracy and ranking. The architectures are represented with operation index per layer.

| Sampling | Arch. | Best Acc. | Avg. Acc. | Best Rank | Avg. Rank |
|---|---|---|---|---|---|
| Boltzmann | [12220111] | 92.39 | $92.30 \pm 0.10$ | 406 | 453 |
| Independent | [22120211] | 92.44 | $92.29 \pm 0.21$ | 347 | 412 |
| MCTS | [22221210] | 92.74 | $92.51 \pm 0.18$ | 80 | 246 |
| Uniform | [21222220] | 92.79 | $92.58 \pm 0.20$ | 56 | 197 |
| MCTS + Reg. (Su et al., 2021a) | [12222222] | 92.92 | $92.67 \pm 0.18$ | 21 | 112 |
| MCTS + Learned (ours) | [22212220] | 93.13 | $92.97 \pm 0.12$ | 1 | 6 |
| Best | [22212220] | 93.13 | - | 1 | - |

Table 4: **Comparison of accuracy and computational cost on ImageNet classification task.** The architecture are searched on Mobilenet-based search space. We consider light weight models with target budget of 280 (top section) and 330 (middle section) MFLOPs. In each section top part is taken directly from the references and the bottom is results of our baselines and our approach. In the bottom section we report other NAS methods outside the budget.

| Method | Best Acc. | FLOPs (M) | Params. (M) | GPU days |
|---|---|---|---|---|
| MobileNetV2(Sandler et al., 2018) | 72.0 | 300 | 3.4 | - |
| MnasNet-A1(Tan et al., 2019) | 75.2 | 312 | 3.9 | 288 |
| SCARLET-C(Chu et al., 2021a) | 75.6 | 280 | 6.0 | 10 |
| OFA (Cai et al., 2019) | 76.0 | 230 | - | $\sim 50$ |
| GreedyNAS-C(You et al., 2020) | 76.2 | 284 | 4.7 | 7 |
| MTC_NAS-C(Su et al., 2021a) | 76.3 | 280 | 4.9 | 12 |
| BigNAS-S (Yu et al., 2020) | 76.5 | 242 | - | $\sim 50$ |
| Uniform | 72.2 | 277 | 4.6 | $\sim 5$ |
| Boltzmann | 73.1 | 278 | 4.7 | $\sim 5$ |
| MCTS + Reg. | 76.0 | 280 | 4.9 | $> 12$ |
| Ours | 76.7 | 280 | 4.9 | $\sim 7$ |
| Proxyless-R (Cai et al., 2018) | 74.6 | 320 | 4.0 | 15 |
| DYNAS-FairNAS (Jeon et al., 2025) | 75.8 | 326 | 4.1 | 16.5 |
| DYNAS-SPOS (Jeon et al., 2025) | 75.9 | 329 | 3.9 | 6.6 |
| DYNAS-Few-Shot (Jeon et al., 2025) | 76.0 | 324 | 4.0 | 31 |
| SPOS (Guo et al., 2020) | 76.2 | 328 | - | 13 |
| SCARLET-B (Chu et al., 2021a) | 76.3 | 326 | 5.2 | 22 |
| FairNAS-C (Chu et al., 2021b) | 76.7 | 325 | 5.6 | $\sim 11$ |
| MCTS_NAS-B (Su et al., 2021a) | 76.9 | 330 | 6.3 | 12 |
| Uniform | 73.1 | 319 | 4.8 | $\sim 6$ |
| Boltzmann | 73.8 | 330 | 6.3 | $\sim 5$ |
| MCTS + Reg. | 76.8 | 330 | 6.3 | $> 12$ |
| Ours | 77.4 | 330 | 6.2 | $\sim 8$ |
| CARS-A (Yang et al., 2020) | 72.1 | 430 | 4.2 | 0.4 |
| ZiCO (Li et al., 2023) | 75.8 | 488 | - | 0.4 |
| MOTE-NAS (Zhang et al., 2024c) | 76.2 | 387 | - | 0.1 |
| GM + DARTS (Hu et al., 2022) | 75.5 | 574 | 5.1 | 2.7 |
| GM + Proxyless (Hu et al., 2022) | 76.6 | 583 | 5.2 | 24.9 |
| PC-DARTS (Xu et al., 2019) | 75.8 | 597 | 5.3 | 3.8 |
| DARTS+ (Liang et al., 2019) | 76.1 | 582 | 5.1 | 6.8 |
| PA&DA (Lu et al., 2023) | 77.3 | 399 | 5.3 | $\sim 9$ |
| OFA (Cai et al., 2019) | 77.7 | 406 | - | $\sim 50$ |

# 6    Conclusion

In this work, we have introduced a novel method to design a hierarchical search space for NAS. We have highlighted the shortcomings of node independence assumption used in popular NAS methods and the impact of the hierarchical search space design on search quality and efficiency. We have shown that by simply learning an appropriate hierarchical representation of the architectures in the search space, we achieve state-of-the-art results with MCTS, without requiring any other form of regularization and with a reduced amount of training. In future work, we will investigate other and more efficient ways to build our hierarchy, focusing in particular to zero cost approaches, which seem promising but still inferior to the used representation.

**Limitations**    Due to the agglomerative clustering, the complexity of our method is quadratic in the number of architectures. However, for the considered setting, the clustering cost is largely dominated by the model inference needed to extract the output representations. We provide a complexity analysis in appendix C.5. For large search spaces the main bottleneck of our method is the memory requirements for distance calculation and clustering. Several optimization techniques reduce these requirements. We applied a FLOPs budget to prune the search space and sample only within a limited range of the budget. Clustering can be made linear instead of quadratic in the number of architectures, by using more efficient clustering methods such as BIRCH (Zhang et al., 1996) or divisive clustering methods such as bisecting k-means (Ran et al., 2023). Distance calculation can be addressed by calculating in blocks and using lower precision to allow it to fit into memory, as well as dimensionality reduction on the output vectors. Finally, the tree can be potentially built dynamically by expanding promising nodes which removes the memory bottleneck.

**Broader Impact Statement**    Our method focuses on automating the search space design and increasing the efficiency of MCTS for one-shot NAS. Besides contribution to the general goal of NAS that aims to democratize AI application by automating a part of the pipeline, our approach reduces the computational cost of one-shot NAS resulting in lower energy consumption and carbon footprint. However, the overall cost of one-shot NAS approaches including ours remains generally high.

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

## A    More Related Work

**Search space design**    Common NAS search spaces can be categorized as Micro (cell-based), Macro, and mulit-level. Micro search spaces (Zoph et al., 2018; Liu et al., 2018b) focus on yielding the optimal architecture by finding the operations that can produce the best model inside a cell or block. The cell is then stacked to form the entire network, while the outer skeleton of the network is often controlled manually by including reduction cells. This was inspired by observing that the state-of-the-art manual architectures were formed by repetition of a certain structure and helped to reduce the complexity of search space to a manageable level (White et al., 2023).

Therefore, the objective of this approach is to find a cell that works well on all parts of the network, which might be suboptimal. This neglects to explore non-homogeneous architectures and diminishes the capability of NAS to find novel architectures. Macro search space (Su et al., 2021a) instead searches for the outer skeleton of the network while fixing the operations at micro level. This can include architecture parameters such as: type of layers, number of layers, or channels in layers, pooling positions etc. Finally, a mulit-level search space searches at two levels: cell and macro structure for a CNN (Liu et al., 2019) or convolution and self-attention layers for vision transformers (Chen et al., 2021a). In this work, we focus on macro search spaces as it is generally more expressive and challenging than micro search spaces.

**NAS benchmarks**    NAS benchmarks are used to evaluate for a given method the quality and the amount of computation required to yield a solution. They have played a crucial role in the NAS community as they provide an evaluation of all architectures in a brute-force way to find the optimal solution and eliminate the need to run independently this expensive process. The development of NAS benchmarks has also improved the reproducibility and efficiency of NAS research. Tabular benchmarks (Ying et al., 2019; Su et al., 2021a) are constructed by exhaustively training and evaluating (metrics such as accuracy, FLOPS, number of parameters, etc.) all possible architectures. On the other hand, surrogate benchmarks (Siems et al., 2020) estimate the architecture performance using a model which is trained on data from several trained architectures. Benchmarks have been developed for both micro (Ying et al., 2019; Dong & Yang, 2020; Siems et al., 2020) (with addition of channels (Dong et al., 2021)) and macro (Su et al., 2021a; Roshtkhari et al., 2023) search spaces. For more details on NAS benchmarks see the survey (Chitty-Venkata et al., 2023).

**Architecture encoding**    Some works have shown that architecture encoding can affect the performance of NAS (White et al., 2020; Ying et al., 2019) and good encoding of architectures enables efficient calculation of relationships or distances among architectures. The most common encoding represents the architecture as a directed acyclic graph (DAG) and adjacency matrix along with a list of operations (Ying et al., 2019; Zoph & Le, 2016). For using performance predictors, BANANAS (White et al., 2021a) proposes a path-based encoding instead of adjacency matrix and GATES (Ning et al., 2020) proposed a graph based encoding scheme that better mode the flow information in the network. Encoding can also be learned by unsupervised training prior to NAS often utilizing an autoencoder (Li et al., 2020b; Lukasik et al., 2021; 2022; Yan et al., 2020; Zhang et al., 2019) or a transformer (Yan et al., 2021). In our work, we make use and compare different ways of encoding architectures for our approach as ablation and show that measuring distances based on the network output seems to be the fundamental for good results.

## B    Experimental Setup and Details

**Sampling method details**    A summary of various methods used in our experiments (Tables 1 and 3) is presented in Table 5.

**Tree design.** Consider the toy search space shown in Fig.2(a), with 3 binary operations $(O_a, O_b, O_c)$ which leads to 8 distinct architectures. The default tree design (Fig. 2 (b)) as presented in sec.3, branches off the tree on operation choices per layer.

Table 5: **Summary of sampling methods used in our experiments.**

| Method | Search Space Structure | Sampling Method |
|---|---|---|
| Uniform | Flat | Uniform sampling of architectures |
| Independent | Flat | Nodes sampled independently |
| Boltzmann | Flat | Joint sampling of architectures |
| Mixture | Flat (partitioned) | Uniform with multiple models (Roshtkhari et al., 2023) |
| MCTS | Hierarchical (def. tree) | Conditional prob. ( sec. 3, Tree Search) |
| MCTS + Reg. | Hierarchical (def. tree) | Conditional prob. + regularization (Su et al., 2021a) |
| MCTS + Learned | Hierarchical (learned tree) | Conditional prob. with learned tree (sec. 4) |

### B.1 Dataset and Hyperparameters

For experiments performed on CIFAR10 (Kocsis & Szepesvári, 2006) dataset, we split the training set 50/50 for NAS training and validation. To tune hyperparameters, we either performed grid search or when comparing with other works used similar hyperparameters. We used SGD with weight decay and cosine annealing learning rate schedule. In all experiments we use $\beta = 0.95$ and $\lambda = 0.5$. Unless specified otherwise, we used L2 distances metric on normalized logits to construct the distance matrix. We used a single random validation batch to obtain the outputs for distance calculation. We used validation batch size of 1024 for CIFAR10 and 256 for ImageNet. We used average clustering linkage in our experiments.

**Pooling search space**  This search space introduced by (Roshtkhari et al., 2023) is based on Resnet20 (He et al., 2016) architecture. The only CNN parameters to search is where to perform pooling. To calculate distance matrix we trained the supernet for 300 epoch using uniform sampling with batch size 512, learning rate 0.1 and weight decay 1e-3. For search we trained for 400 epochs with batch size 256, learning rate learning rate 0.05 and temperature $T$ is set to linearly annealing schedule (0.02, 0.0025). Since this search space is small we only consider nodes with max probabilities and report it as the final architecture. For this search space we utilized KL distance.

**NAS-Bench-Macro search space**  This benchmark introduced by (Su et al., 2021a) is based on MobileNetV2 (Sandler et al., 2018) blocks. The supernet warm-up is performed for 80 epochs with batch size 512 and learning rate 0.05. For search we use batch size of 256 for 120 epochs with and $T$ linearly annealing from 0.01. At the ends of training, we sample 50 architectures from the tree and report the best as final architecture.

**ImageNet**  To accelerate our training in ImageNet experiments, we use mixed precision and FFCV (Leclerc et al., 2023) library. We sample architectures within a FLOPs budget and discard those outside of it. We train for 100 epochs with SGD and cosine annealing learning rate. Other training strategies are similar to experiments on CIFAR10.

## C  Additional Results

### C.1  Reward for MCTS

The most common rewards used for NAS algorithms is accuracy and loss. While loss is differentiable, accuracy is more aligned with the objective of NAS. Furthermore, either the training or validation can be used to calculate the reward. Instead of absolute values, a relative training loss metric was used in (Su et al., 2021a) to account for unfair reward comparison at different iteration of supernet training. In Table 6 for NAS-Bench-Macro, we explore some common combination of options to estimate the reward. In all setting our approach performs on par or slightly better than Default Tree + Regularization. For both methods it seems that using the accuracy on the validation set as metric is the best. However, while for our approach the best performing configuration is obtained with the absolute metric, for Su et al. (2021a) the relative metric seems slightly better.

Table 6: **CIFAR10 results on NAS-Bench-Macro (Su et al., 2021a)** search space with several various rewards. Relative rewards are calculated according to Su et al. (2021a). The rewards can be can be calculated on either training or validation set.

| Search Structure | Metric | Reward Data | Reward Measure | Arch. | Best Acc. | Best Rank | Avg. Rank |
|---|---|---|---|---|---|---|---|
| Default Tree + Reg | rel. | train | loss | [22121222] | 92.74 | 85 | 97 |
| Learned Tree (ours) | rel. | train | loss | [22122220] | 92.78 | 61 | 67 |
| Default Tree + Reg | abs. | train | acc. | [22121210] | 92.55 | 227 | 278 |
| Learned Tree (ours) | abs. | train | acc. | [22110222] | 92.56 | 209 | 301 |
| Default Tree + Reg | rel. | val. | loss | [22222022] | 92.71 | 98 | 120 |
| Learned Tree (ours) | rel. | val. | loss | [21211220] | 92.76 | 71 | 95 |
| Default Tree + Reg | rel. | val. | acc. | [12222222] | 92.92 | 21 | 112 |
| Learned Tree (ours) | rel. | val. | acc. | [22212200] | 92.94 | 19 | 67 |
| Default Tree + Reg | abs. | val. | acc. | [22221200] | 92.86 | 34 | 54 |
| Learned Tree (ours) | abs. | val. | acc. | [22212220] | 93.13 | 1 | 6 |

Table 7: **Distance measures for the similarity matrix.** We compare the final performance of our MCTS on Pooling benchmark with the learned tree structure which is built with an agglomerative clustering using a similarity matrix between network outputs with different distance measures.

| Distance Measure | Best Arch. | Best Acc. | Avg, Acc. |
|---|---|---|---|
| cross-entropy | [5,3,2] | 91.55 | $91.20 \pm 0.23$ |
| L2 | [6,2,2] | 91.83 | $91.52 \pm 0.16$ |
| KL | [6,2,2] | 91.83 | $91.72 \pm 0.12$ |

## C.2 More Ablations

**Distance metric**   Using CNN's output to calculate pairwise distances, the difference between two distributions can be calculated by several distance measures. In table 7, we investigate three common distance measures: L2 distance, KL divergence, and cross-entropy.

**Smoothing factor**   We provide sensitivity analysis for parameter $\beta$ in eq. 4 in fig. 5. In single path training one architecture is sampled and the reward is based on one mini-batch of data, making them noisy. Parameter $\beta$ controls the noise and how quickly MCTS adapts to new data. Therefore, smaller values result in noisy rewards and convergence to suboptimal architectures. However, over-smoothing ($\beta = 0.99$) leads to slightly worse performance due to slower adaptation to new information, resulting in prolonged effect of historical reward values and potentially slower convergence.

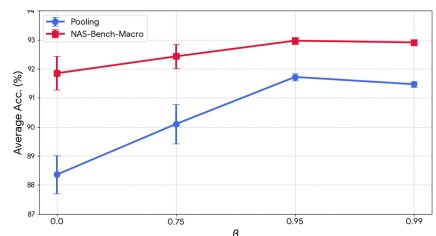

Figure 5: **Ablation on smoothing factor.** We compare the average accuracy of MCTS with various smoothing factors.

**Warm-up iterations**   During supernet training, the average accuracy typically increases. In general, since the criteria for our proposed clustering is similarity and not the exact accuracy, the supernet needs to be trained only long enough to capture that similarity. To investigate the impact of warm-up duration, we evaluated MCTS with the hierarchy built at various iterations for Pooling benchmark (Fig. 6.(b)). The performance shows significant

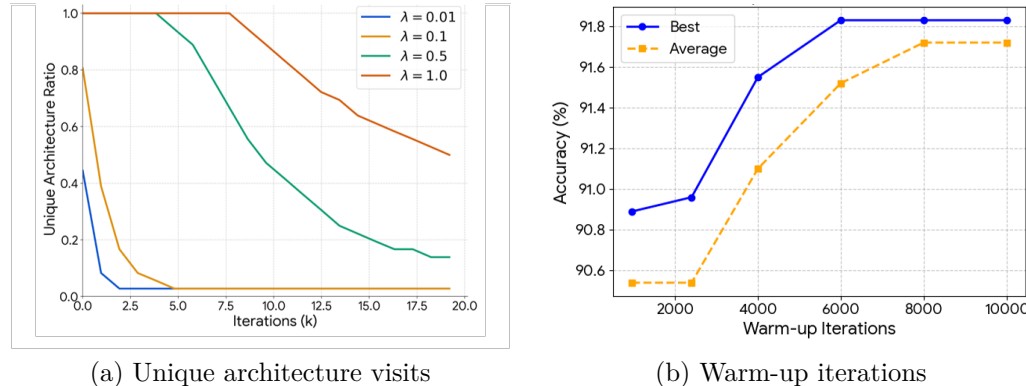

(a) Unique architecture visits      (b) Warm-up iterations

Figure 6: **(a) Unique architecture visits.** We show number of unique architecture visited during training. **(b) Warm-up iterations for estimating the distance matrix.** We show the final performance of our MCTS in which the tree structure is learned with a model uniformly trained for a given number of iteration.

gains during the training of supernet and eventually plateaus as the tree structure stabilize. Therefore, while sufficient warm-up is required for the best performance, the tree structure does not meaningfully change after a certain point.

**Exploration parameter** In table 8 we examine the sensitivity of the method to the exploration parameter $\lambda$ in eq. 3. Parameter $\lambda$ controls the exploration-exploitation trade-off. $\lambda = 0$ is the greedy search with pure exploitation, while $\lambda > 2$ shows more exploration and similar results to uniform sampling. Figure 6.(a) shows the percentage of unique architectures sampled at different stages of MCTS. It shows the exploration-exploitation balance; when exploring a variety of different architectures are sampled while exploitation focuses on a narrow subset of architectures. With small $\lambda$ (0.01, 0.1) the sampling quickly collapses to a narrow subset of architectures, while larger values (0.5, 1.0) promote more exploration and sampling a wider set of architectures.

Table 8: **Ablation on exploration factor.** We report variance in average distances and the accuracies for three validation batches.

| $\lambda$ | Pooling | NAS-Bench-Macro |
|---|---|---|
| 0.01 | $89.08 \pm 1.23$ | $89.27 \pm 1.53$ |
| 0.5 | $91.72 \pm 0.12$ | $92.97 \pm 0.12$ |
| 1.0 | $91.10 \pm 0.54$ | $92.91 \pm 0.14$ |
| 2.0 | $90.65 \pm 0.22$ | $92.69 \pm 0.18$ |

**Clustering Linkage** We compare different linkage criteria for agglomerative clustering in table 9. We report average accuracy and Cophenetic Correlation Coefficient (CPCC) which measures how the dendrogram preserves the pairwise distances. Single linkage returns the minimum distance between two points and can be sensitive to outliers. Complete linkage merges clusters based on the maximum distances. Average linkage can be seen as a compromise between the two, using average distance of all pairs. Ward's linkage tries to minimize the variance within the cluster during merging.

Table 9: **Ablation on clustering linkage criteria.** We report accuracy and Cophenetic Correlation Coefficient (CPCC) for various linkages in Pooling search space.

| Linkage | Avg. Acc. | CPCC |
|---|---|---|
| Single | $91.28 \pm 0.39$ | 0.41 |
| Complete | $91.43 \pm 0.34$ | 0.47 |
| Average | $91.72 \pm 0.12$ | 0.73 |
| Ward | $91.78 \pm 0.18$ | 0.67 |

**Validation batch** Since our hierarchy is based on the output of a validation batch of data, we analyze the sensitivity to the input batch and the stability of the . We chose 3 random batches of data (512 or 1024 images each) and calculated the variance in distance matrix. Since architectures accuracy for CIFAR10 is relatively high, many models produce an output vector with values close to one. Therefore, to focus on

Table 10: **Ablation on validation batch.** We report variance in average distances and the accuracies for three validation batches.

| | Pooling | | NAS-Bench-Macro | |
|---|---|---|---|---|
| Validation Batch | Distance Var. | Avg. Acc. | Distance Var. | Avg. Acc. |
| Random Batch (512) | 0.17 | $91.46 \pm 0.30$ | 0.11 | $92.94 \pm 0.12$ |
| Random Batch (1024) | 0.12 | $91.72 \pm 0.12$ | 0.05 | $92.97 \pm 0.12$ |
| High Variance | 0.21 | $91.02 \pm 0.95$ | 0.18 | $92.37 \pm 0.93$ |

Table 11: **Ablation on temperature.** We report average performance with various temperature schedules.

| Schedule | Pooling | NAS-Bench-Macro |
|---|---|---|
| Constant | $91.00 \pm 0.03$ | $91.70 \pm 0.77$ |
| Linear | $91.72 \pm 0.12$ | $92.97 \pm 0.12$ |
| Exponential | $91.36 \pm 0.62$ | $92.68 \pm 0.15$ |

architecture disagreements and provide better contrast among architectures, we sampled 9 random batches of images. We calculated the variance in outputs across those images and kept 3 batch of images with the highest variance. These batches represent a more discriminative set of samples for architectures. We compare the average variance and average accuracy of architectures in tab. 10.

**Temperature scheduling**   We consider linear, exponential and constant temperature $T$ in table 11.

## C.3   Distance Matrices

We visualize the distance matrices calculated using output vector and various encoding in Fig. 7.

## C.4   Tree Visualizations

For pooling search space, we visualize tree structure based on our proposed method and architecture encodings in Fig. 8. The tree is presented with architecture indices on leaves. The architecture indices and the corresponding performance can be found in Roshtkhari et al. (2023).

## C.5   Complexity Analysis

Considering $N$ architectures in the search space, the computational complexity to build the tree for our method is determined by two factors: the complexity of inference to obtain output vectors from the supernet after warm-up ($O(N)$) and the cost of distance calculation and clustering ($O(N^2)$). The cost of inference depends on the complexity of the architectures in the search space, while the output distance calculation depends on number of output classes. Therefore, the total complexity can be estimated as $aN^2 + bN$. We estimate that our method works best for $N < 10k$ and estimate values of $a = 1e - 8$ and $b = 0.01$ for our ImageNet experiments. Therefore, the cost of inference dominates the overall cost.

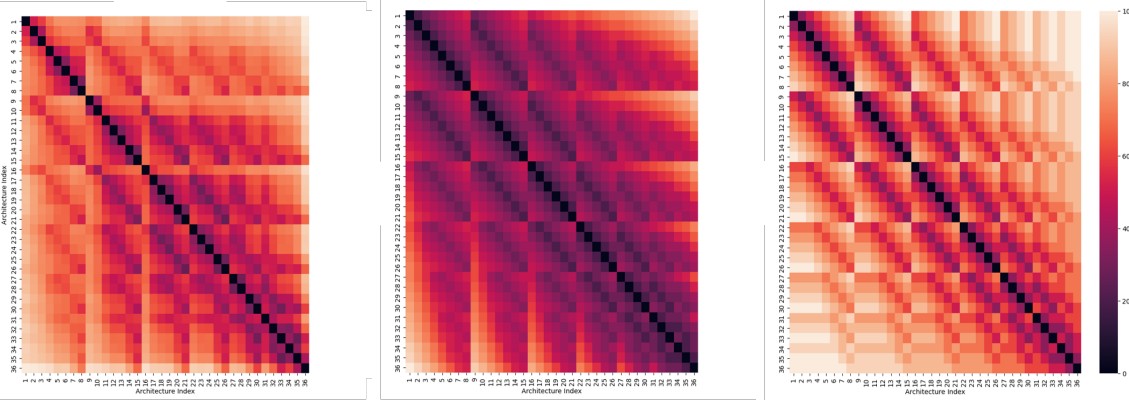

Figure 7: **Normalized distance matrices calculated with various methods.** (left) Distance matrix calculated from output vectors (our method) ; (middle) From vector encoding ; (right) From one-hot encoding. The architecture indices on leaves correspond to indices used in Pooling benchmark (Roshtkhari et al., 2023).

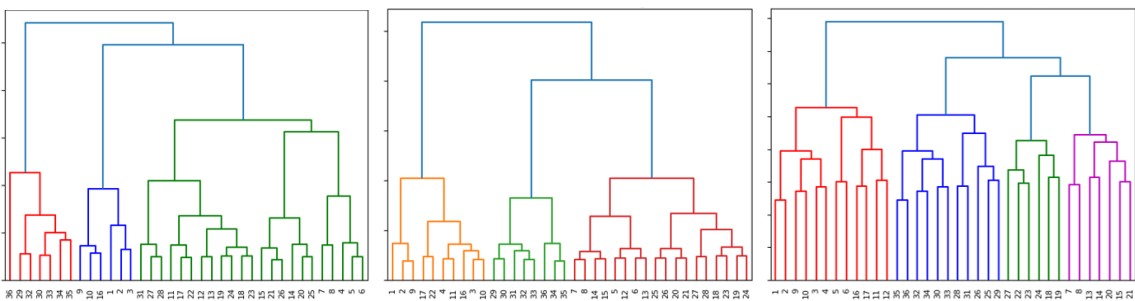

Figure 8: **Tree branching for Pooling search space by hierarchical clustering.** The architecture indices on leaves correspond to indices used in Pooling benchmark (Roshtkhari et al., 2023) (left) Tree learned from output vectors (our method) ; (middle) From vector encoding ; (right) From one-hot encoding.

