# OpenReview forum: "Neural Architecture Search by Learning a Hierarchical Search Space"
_TMLR — Rejected by TMLR_

### Review · Reviewer_fPny · 2025-12-19

**Summary Of Contributions:**

This paper studies how to make Monte-Carlo Tree Search (MCTS) effective for one-shot Neural Architecture Search (NAS) when the search space is discrete and large, and where the “order” of branching decisions is not intrinsically fixed (unlike games). The core idea is to learn a better hierarchical factorization of the architecture distribution by constructing a search tree whose internal nodes group “similar” architectures, so that early branching decisions separate semantically meaningful subsets and improve sampling efficiency. The learned hierarchy improves performance and search efficiency versus uniform, Boltzmann, independent-node sampling, and MCTS with a default (layer-wise) tree, and can match or improve over MCTS+regularization baselines without requiring the same regularization trick.

**Audience:**

Yes

**Audience Explanation:**

TMLR readers interested in AutoML, NAS, and black-box/discrete optimization should find this work relevant. The contribution sits at the intersection of (i) sampling and probability factorization in one-shot NAS, and (ii) improved search efficiency for MCTS through learned hierarchical structure. The paper’s message that “tree design is a first-class object” for MCTS-based NAS, and that functional similarity (via output vectors) can provide a strong inductive bias for that design, seems broadly useful beyond the specific benchmarks presented.

**Claims And Evidence:**

Yes

**Claims Explanation:**

For the main claims (that a learned hierarchy improves MCTS-based NAS training and search efficiency, and that output-space clustering is a better basis for partitioning than supernet-accuracy partitioning), the paper provides direct comparative experiments and ablations.

- On the Pooling CIFAR-10 benchmark, the learned-tree method improves best and average accuracy and ranking relative to MCTS baselines, with comparable GPU hours.
- The ablation comparing accuracy-based partitioning versus output clustering indicates the former is “notably worse,” supporting the argument that supernet accuracy estimates can be unreliable for building the hierarchy.
- On ImageNet, the method shows improved accuracy compared to the paper’s sampling baselines and favorable compute compared to MCTS+regularization, while also being competitive with prior NAS methods reported in the table.

**Requested Changes:**

1. The algorithmic description suggests computing outputs and pairwise distances for architectures in the search space, which is infeasible when the number of possible architectures is enormous. Please explicitly describe what subset is clustered in the ImageNet setting (or what constraints reduce the effective N), and provide the exact procedure used in practice. Also report the number of architectures actually clustered and the memory/time cost for building the distance matrix and clustering. This is central given the quadratic cost discussed.

2. Specify the exact distance metric on outputs, how outputs are formed (which validation batch size, number of batches, concatenation details), clustering linkage/criterion, and any normalization. These choices can materially affect the hierarchy.

3. Please include more sensitivity analysis such as warm-up length, choice of validation batch(es) for output vectors, clustering linkage, and temperature schedule to show robustness.

---

> ### Author Response · Authors · 2026-02-15
> **Response to Reviewer fPny**
>
> We thank the reviewer for their valuable feedback and suggestions. We are thrilled that the reviewer found our work interesting and our claims well supported. Below we address the requested changes by the reviewer:
>
> > 1. The algorithmic description suggests computing outputs and pairwise distances for architectures in the search space, which is infeasible when the number of possible architectures is enormous. Please explicitly describe what subset is clustered in the ImageNet setting (or what constraints reduce the effective N), and provide the exact procedure used in practice.
>
> We used a FLOPs reduction technique to prune the search space (discussed in section 5.4). Given the target FLOPs budget (280 and 330 MFLOPs) we used the FLOPs of architectures as a very simple form of zero-cost proxy. We only accepted sampled architectures within close to the budget ([0.99,1] x budget) which is a similar technique used in Su et al. (2021a).
>
> > ... Also report the number of architectures actually clustered and the memory/time cost for building the distance matrix and clustering. This is central given the quadratic cost discussed.
>
> We clustered 10k architectures for the imagenet. Please note that we have incorporated the time cost of building the distance matrix and clustering in the time we report in table 4.
>
> We estimate the total time cost of evaluation and building the distance matrix and clustering as ~4 hours using 40 GB memory .The main bottleneck is the memory for distance matrix calculation, which can be partially addressed by calculating in blocks and using lower precision to allow it to fit into memory. We edited the “Limitations” section to clarify the bottleneck.
>
> > 2. Specify the exact distance metric on outputs, how outputs are formed (which validation batch size, number of batches, concatenation details), clustering linkage/criterion, and any normalization. These choices can materially affect the hierarchy.
>
> In Appendix B.1 we added the following specific choices for our experiments:
> * Distance used for outputs. We also include an ablation on three distance choices (cross-entropy, L2,  and symmetric KL) for the pooling search space in table 7.
> * How outputs are formed: batch size, number of batches and selection
> * Clustering linkage used
>
> > 3. Please include more sensitivity analysis such as warm-up length, choice of validation batch(es) for output vectors, clustering linkage, and temperature schedule to show robustness.
>
> We added requested ablations in supplementary C.2 including:
> * Warm-up length (Fig. 6 (b))
> * Choice of validation batch (Tab. 10)
> * Clustering linkage (Tab. 9)
> * Temperature scheduling (Tab.11)
>
> Additional ablations added:
> * Smoothing factor $\beta$ (Fig. 5)
> * Exploration parameter $\lambda$ (Tab. 8 and Fig. 6(a))

---

### Review · Reviewer_Yc5X · 2026-02-02

**Summary Of Contributions:**

The authors proposed a novel NAS methodology which aims at optimizing the hierarchical search of architectures by defining the problem as a Monte-Carlo tree search optimization where similarly behaving architectures are clustered together. The proposed methodology is tested over a couple of scenarios showcasing promising performance. However, the novelty of the proposed approach is quite limited and the experimental evaluation is rather narrow and lacks comparison with several important approaches in the field.

**Additional Comments:**

- The authors should thoroughly proofread the manuscript. There are multiple grammatical errors and typos, including missing brackets (e.g., *“(see Fig. 4”*, incorrect or awkward expressions *“A simplest way”*, and subject‑verb agreement mistakes *“The root node of the tree correspond”* instead of *“corresponds”*).
- I recommend adding a discussion in the related work about NAS methods that generalize across datasets without retraining a search procedure, particularly those based on performance transfer and architecture generation. These works could also serve as additional baselines for low-resource NAS.
- It would be helpful if the authors provided more intuition or examples clarifying the practical process behind their assumptions (e.g., how to obtain the fully pre-expanded tree in practice, how the similarity metric behaves in large spaces, etc.).
- While the method may work well on small or toy search spaces, the paper would benefit from a more transparent discussion about its limitations in realistic NAS settings, especially given that modern architectures tend to require large, expressive search spaces with many possible operations and hyperparameters.
- The authors may want to revisit the discussion of their complexity analysis, as the limitation of N < 10k appears central to the applicability of the entire method.

**Audience:**

Yes

**Audience Explanation:**

The authors tackle a relevant challenge of current ML systems, which heavily rely on NNs and for which the architecture design may largely impact the performance of the system. Therefore, I believe that the paper would be of interest to at least a part of the TMLR’s audience.

**Broader Impact Concerns:**

The authors did not onclude a broader impact concern section. While, I understand that the topic is rather self-contained and that the authors are focusing mostly on the research perspective, I believe that the authors should provide an additional section where they discuss the potential impact of their work on the real world and what could be the possible issues or challenges deriving from its application.

**Claims And Evidence:**

No

**Claims Explanation:**

The authors analyse the performance of the proposed methodology over a few scenarios and against a handful of other NAS approaches. Therefore, it seems that the provided methodology has potential. However, this experimental evaluation is rather limited, as it does not include a large portion of recently proposed very effective NAS methods as well as standard benchmarks. Therefore, the claims made about the effectiveness of the proposed approach can be backed only partially.

**Requested Changes:**

**Critical:**
- The contribution is fairly narrow, as the authors essentially extend the definition of MCTS applied to NAS to make it independent of NN layers and rely on architecture similarity to boost MCTS performance. While potentially effective, this represents only a small incremental improvement over well-established approaches. This raises concerns about whether the work meets the novelty threshold for publication.
- The assumption that the search tree is fully pre‑expanded is not convincingly justified. As stated, this would require a pre-trained and reliable supernet in which both model and supernet weights are already known. In practice, this appears infeasible, since it would essentially require training an extremely large number of architectures for a sufficiently large search space. The authors should substantially clarify the feasibility of this assumption and add a more detailed discussion of the computational requirements and practical limitations of their approach.
- Equation 4 appears incorrect or misleading. It can be easily re-written as $C(a_i)$ $\cdot$ (1- $\beta$) = (1 - $\beta$) $\cdot$ Acc, which means that $C(a_i)$ = Acc, meaning that no smoothing happens and $C(a_i)$ is just the validation accuracy. Either Equation 4 is incorrect/mistyped, or it is intentionally masking the simplicity of the computation. This issue needs to be clarified explicitly.
- Experimental evaluation is too limited in scope. The evaluation does not include standard NAS benchmarks such as NAS-bench-101 [1], Nas-bench-201 [2], NATS-Bench [3] and more [4]. These are widely considered the de-facto golden standard for NAS evaluation, and omitting them raises doubts about the generalizability of the claimed results. The chosen scenarios are very limited (e.g., CIFAR-10 with only 36 architectures), which makes it difficult to assess the true effectiveness of the approach. The authors should significantly broaden the evaluation benchmarks.
- Missing comparisons with state‑of‑the‑art NAS methods. The proposed method is compared only against relatively weak baselines. State-of-the-art approaches such as NPENAS [5], PC-DARTS [6], BANANAS [7], PRT [8], and more are missing. A solid comparison with these methods is needed to properly position the contribution.
- The statement *“we estimate that our method works best for N < 10k”* is a significant drawback, as many modern NAS search spaces are much larger. This limitation directly questions the applicability of the approach in real-world settings. The authors should discuss this drawback in depth and explain how the approach could be used or extended for realistic, large-scale NAS.
- The proposed method appears to suffer from the same issue affecting many NAS approaches: poor generalization across tasks. The search tree construction and splitting process must be restarted for each new dataset or task. This is a substantial limitation, especially compared to training-free or generalizable NAS methods ([9,10]). The authors should discuss this problem explicitly and consider how their approach could improve generalization across tasks.

**Would Strengthen the Work:**
- Several relevant works on generalizable or training-free NAS (e.g., [9,10]) are omitted. Including these as related work and possibly as baselines would strengthen the positioning of the paper.
- Beyond addressing the feasibility of the pre-expansion assumption, a general paragraph describing computational cost, scalability, and practical constraints would improve the clarity of the contribution.

[1]. Ying, Chris, et al. "Nas-bench-101: Towards reproducible neural architecture search." International conference on machine learning. PMLR, 2019.

[2]. Dong, Xuanyi, and Yi Yang. "Nas-bench-201: Extending the scope of reproducible neural architecture search." arXiv preprint arXiv:2001.00326 (2020).

[3]. X. Dong, L. Liu, K. Musial and B. Gabrys, "NATS-Bench: Benchmarking NAS Algorithms for Architecture Topology and Size," in IEEE Transactions on Pattern Analysis and Machine Intelligence, vol. 44, no. 7, pp. 3634-3646, 1 July 2022, doi: 10.1109/TPAMI.2021.3054824.

[4]. Mehta, Yash, et al. "NAS-bench-suite: NAS evaluation is (now) surprisingly easy." arXiv preprint arXiv:2201.13396 (2022).

[5]. Wei, Chen, et al. "Npenas: Neural predictor guided evolution for neural architecture search." IEEE transactions on neural networks and learning systems 34.11 (2022): 8441-8455.

[6]. Xu, Yuhui, et al. "Pc-darts: Partial channel connections for memory-efficient architecture search." arXiv preprint arXiv:1907.05737 (2019).

[7]. White, Colin, Willie Neiswanger, and Yash Savani. "Bananas: Bayesian optimization with neural architectures for neural architecture search." Proceedings of the AAAI conference on artificial intelligence. Vol. 35. No. 12. 2021.

[8]. Li, Nan, et al. "Transferable Relativistic Predictor: Mitigating Cross-Task Cold-Start Issue in NAS." Proceedings of the Thirty-Fourth International Joint Conference on Artificial Intelligence. 2025.

[9]. Agiollo, Andrea, and Andrea Omicini. "GNN2GNN: Graph neural networks to generate neural networks." Uncertainty in Artificial Intelligence. PMLR, 2022.

[10]. Lee, Hayeon, Eunyoung Hyung, and Sung Ju Hwang. "Rapid neural architecture search by learning to generate graphs from datasets." arXiv preprint arXiv:2107.00860 (2021).

---

> ### Author Response · Authors · 2026-02-15
> **Response to Reviewer Yc5X (part 1)**
>
> We thank the reviewer for their insightful review and comments. Below we address the requested changes:
>
> > The contribution is fairly narrow, as the authors essentially extend the definition of MCTS applied to NAS to make it independent of NN layers and rely on architecture similarity to boost MCTS performance. While potentially effective, this represents only a small incremental improvement over well-established approaches. This raises concerns about whether the work meets the novelty threshold for publication.
>
> The main idea of the paper is that while probabilistically the orderings of a chain of conditional probabilities does not matter for the final joint distribution, during learning the ordering is very important. This reflects the recent trends in other contexts such as text/image generation shifting to masked diffusion and order-agnostic autoregressive models in which a specific generation order is induced for improved results [11].
>
> In this paper, we implemented this idea for the specific case of MCTS for NAS. Previous methods either used a predefined order (Su et al. (2021a)) or did not learn the supernet and the hierarchy jointly (Wang et al., 2021b, Zhao et al., 2021b and other methods that are not one-shot such as Le et al., 2024).
>
> We propose to change how the search hierarchy is constructed, shifting from the structural or supervised hierarchy to unsupervised functional hierarchy. This unsupervised approach is particularly important when supernet training and search are performed simultaneously (and not in two separate stages), in which the supernet performance evaluations are not reliable to accurately rank and distinguish architectures.
>
> We show that our method saturates the performance on benchmarks, while reducing the search time. While the absolute absolute gains in this benchmarks are limited due to the saturation, we obtained almost optimal results with comparable or less computation. On ImageNet dataset with MobileNet search space, we obtained 77.4 % accuracy comparable to methods such as OFA with lower computational cost.
>
> [11] Learning-Order Autoregressive Models with Application to Molecular Graph Generation, ICLR, 2025
>
> > The assumption that the search tree is fully pre‑expanded is not convincingly justified. As stated, this would require a pre-trained and reliable supernet in which both model and supernet weights are already known. In practice, this appears infeasible, since it would essentially require training an extremely large number of architectures for a sufficiently large search space. The authors should substantially clarify the feasibility of this assumption and add a more detailed discussion of the computational requirements and practical limitations of their approach.
>
> The full expansion does not require the estimation of probabilities of all nodes and is also used in Su et al., 2021a. We need a limited warm-up period of supernet training with uniform sampling to build the tree (Uniform sampling is the standard warm-up used in many papers). After the warm-up the probabilities are estimated along with the training of supernet.
>
> While the performance estimates from the supernet are not very reliable, the distances/similarities provided by the supernet are adequate for building the hierarchy. The full expansion means the hierarchy is established once after the warm-up phase and does not change during the training, however this is not necessarily a requirement for the method, but a simplification similar to Su et al., 2021a.
>
> > Equation 4 appears incorrect or misleading. It can be easily re-written as  $C(a_i) \cdot (1- \beta) = (1 - \beta ) \cdot Acc$, which means that $C(a_i) = Acc$, meaning that no smoothing happens and $C(a_i)$ is just the validation accuracy. Either Equation 4 is incorrect/mistyped, or it is intentionally masking the simplicity of the computation. This issue needs to be clarified explicitly.
>
> We understand how the presentation may have been misunderstood. In fact, Eq. 4 $C(a_i)$ is the smoothed version of $Acc$ with iterative updates. On the left it represents the updated value and the right side is the previous stored value. To increase clarity we replaced the “=” with the assignment notation “$\leftarrow$”.

---

> ### Author Response · Authors · 2026-02-15
> **Response to Reviewer Yc5X (part 2)**
>
> > Experimental evaluation is too limited in scope. The evaluation does not include standard NAS benchmarks such as NAS-bench-101 [1], Nas-bench-201 [2], NATS-Bench [3] and more [4]. These are widely considered the de-facto golden standard for NAS evaluation, and omitting them raises doubts about the generalizability of the claimed results. The chosen scenarios are very limited (e.g., CIFAR-10 with only 36 architectures), which makes it difficult to assess the true effectiveness of the approach. The authors should significantly broaden the evaluation benchmarks.
>
> We chose the search spaces and benchmark carefully to provide small scale challenging spaces and a popular large scale.
> Pooling benchmark is a small but challenging search space where due to the specific full weight sharing, many NAS methods (e.g. differentiable) fail.
>
> NAS-Bench-Macro search space provides comparison and ablation while Mobilenet search space is a standard large scale search space for ImageNet.
> We chose NAS-Bench-Macro and Mobilenet search space to provide the direct comparison to the closest method (Su et al., 2021a).
>
> While we believe the provided experimental results and additional ablations adequately show the effectiveness of our approach, we plan on testing on additional benchmarks. Unfortunately due to the limited time of the rebuttal we do not have additional results at this time. We will try our best to provide additional results by the end of the rebuttal or for the camera-ready version.
>
> > Missing comparisons with state‑of‑the‑art NAS methods. The proposed method is compared only against relatively weak baselines. State-of-the-art approaches such as NPENAS [5], PC-DARTS [6], BANANAS [7], PRT [8], and more are missing. A solid comparison with these methods is needed to properly position the contribution.
>
> We updated table 4 to include comparisons with more NAS methods based on Mobilenet search space. We included various fundamental and recent methods covering various approaches (one-shot, zero-shot, etc.) and compared our method both with architectures within the FLOPs budget and in general. In our comparisons our method provides the second-best performance with lower cost.
>
> > The statement “we estimate that our method works best for N < 10k” is a significant drawback, as many modern NAS search spaces are much larger. This limitation directly questions the applicability of the approach in real-world settings. The authors should discuss this drawback in depth and explain how the approach could be used or extended for realistic, large-scale NAS.
>
> We edited the "Limitations" section to provide more details and clarification.
>
> To clarify the statement the main bottleneck for larger search spaces is the memory requirement for distance matrix calculation and clustering. There are several optimization techniques that can be applied for larger search spaces. For Mobilenet search space, similar to Su et al., 2021a we used FLOPs as a very simple zero-cost proxy to prune the search space and sample architectures within a budget range.
>
> Clustering can be made linear instead of quadratic by using more efficient clustering methods such as BIRCH, and devising methods such as bisecting k-means. Distance calculation can be addressed by calculating in blocks and using lower precision to allow it to fit into memory, as well as dimensionality reduction on the output vectors. Finally, the tree can be potentially built dynamically using a sampling strategy (e.g. controller) and placing the architectures in appropriate branches during the training. which removes the memory bottleneck.
>
> Overall, we proposed a new method and showed its potential in both small and large scale search spaces, while highlighting the limitation at its current form. As mentioned above these limitations can be removed or mitigated, however providing a ready-to-use algorithm that works in all conditions is outside the scope of our current work.
>
> > The proposed method appears to suffer from the same issue affecting many NAS approaches: poor generalization across tasks. The search tree construction and splitting process must be restarted for each new dataset or task. This is a substantial limitation, especially compared to training-free or generalizable NAS methods ([9,10]). The authors should discuss this problem explicitly and consider how their approach could improve generalization across tasks.
>
> We thank the reviewer for this insightful comment. It is true that, like many NAS methods, our current framework constructs the search tree from scratch for a new task. We acknowledge that this is a limitation compared to training-free methods, which prioritize search speed and cross-task generalization, while one-shot methods focus on obtaining the best performance on the given dataset.

---

> ### Author Response · Authors · 2026-02-15
> **Response to Reviewer Yc5X (part 3)**
>
> > Several relevant works on generalizable or training-free NAS (e.g., [9,10]) are omitted. Including these as related work and possibly as baselines would strengthen the positioning of the paper.
>
> We added a section to the "Related work" to include training-free and generalizable NAS approaches. Training-free methods (zero-cost proxies) are very efficient but do not achieve the SOTA performance, while generative methods often require training on a benchmark to estimate the distributions.
>
> > Beyond addressing the feasibility of the pre-expansion assumption, a general paragraph describing computational cost, scalability, and practical constraints would improve the clarity of the contribution.
>
> We included a paragraph at the "Limitations" section to provide these details.
>
> ### **Broader Impact Concerns:**
> > The authors did not onclude a broader impact concern section. While, I understand that the topic is rather self-contained and that the authors are focusing mostly on the research perspective, I believe that the authors should provide an additional section where they discuss the potential impact of their work on the real world and what could be the possible issues or challenges deriving from its application.
>
> We included a "Broader impact statement" at the end of the main manuscript.
>
> ### **Additional Comments:**
> > The authors should thoroughly proofread the manuscript....
>
> We apologize for the grammatical mistakes and typos. We proofread the manuscript and corrected the errors.
>
> > I recommend adding a discussion in the related work about NAS methods that generalize across datasets without retraining a search procedure, particularly those based on performance transfer and architecture generation. These works could also serve as additional baselines for low-resource NAS.
>
> We added a paragraph to the related work to include training-free and generalizable NAS approaches. We also included several works that search on mobilenet space to table 4.
>
> > It would be helpful if the authors provided more intuition or examples clarifying the practical process behind their assumptions (e.g., how to obtain the fully pre-expanded tree in practice, how the similarity metric behaves in large spaces, etc.).
>
> As mentioned in previous parts of the response, in practice, we did not work directly with the entire search space as we used a FLOPs budget to prune the search space (similar to works such as Su et al., 2021a, [12]).
>
> [12] Fast Hardware-Aware Neural Architecture Search, 2020.
>
> > While the method may work well on small or toy search spaces, the paper would benefit from a more transparent discussion about its limitations in realistic NAS settings, especially given that modern architectures tend to require large, expressive search spaces with many possible operations and hyperparameters.
>
> We added clarifications and more details to the "Limitation" section. We note that our method achieves competitive results for ImageNet in Mobilenet search space with our pruning approach. To reiterate, for a very large search, the memory bottleneck for clustering can be reduced with various techniques such as alternative clustering algorithms.
>
> > The authors may want to revisit the discussion of their complexity analysis, as the limitation of N < 10k appears central to the applicability of the entire method.
>
> We added more details in "Limitation" on the specific bottlenecks directly related to search space constraints in the complexity analysis.

---

### Review · Reviewer_feTV · 2026-02-05

**Summary Of Contributions:**

This paper investigates Monte Carlo Tree Search (MCTS) for Neural Architecture Search (NAS). The paper demonstrates that modeling architectures using conditional probabilities within a hierarchical search space can lead to improved solutions compared to overly restrictive assumptions. They further show that using pairwise distances between architectures is the most effective approach for constructing the hierarchy. The experimental results demonstrate the effectiveness of the proposed method.

**Audience:**

Yes

**Audience Explanation:**

NAS is an important topic in the field.

**Broader Impact Concerns:**

No concerns.

**Claims And Evidence:**

Yes

**Claims Explanation:**

Overall, the experimental results are aligned with the stated claims and are convincing within the evaluated settings.

**Requested Changes:**

1. The hierarchy is constructed once using output vectors obtained after the warm-up phase and is kept fixed, even though the supernet weights continue to evolve. It would strengthen the paper to include a discussion of why the authors do not update the output vectors and hierarchy during training, and what the motivation is behind this choice.

2. The choice of the smoothing factor $\beta$ in the reward update is not analyzed. Since $\beta$ directly affects how quickly MCTS adapts to new information versus retaining historical estimates, an ablation on its impact is necessary.

3. Following the second point, the exploration parameter $\lambda$ is also not analyzed. And a corresponding analysis is necessary.

4. The authors argue that the method performs conditional sampling; however, this is never explicitly materialized in the paper. In particular, the numeric form of $p(a_t \mid a_{<t})$ is not specified.

5. The authors should analyze the impact of the chosen sampling strategy and evaluate alternative sampling methods under their framework.

---

> ### Author Response · Authors · 2026-02-15
> **Response to Reviewer feTV**
>
> We thank the reviewer for their valuable feedback and comments. We are delighted that that the reviewer found our work important and our experimental results convincing. We address the requested changes below:
>
> >  1. The hierarchy is constructed once using output vectors obtained after the warm-up phase and is kept fixed, even though the supernet weights continue to evolve. It would strengthen the paper to include a discussion of why the authors do not update the output vectors and hierarchy during training, and what the motivation is behind this choice.
>
> We chose to construct the hierarchy only once as it is a more straightforward approach and saves some computation in our setting.
> While supernet weights continue to get refined, the relative similarity/distance metric we use remains mostly stable. We determined that the outputs adequately capture the similarities after the warm-up phase.
> To analyze whether the hierarchy constructed at various points during supernet training is meaningfully different, we performed sensitivity analysis for the warm-up phase iteration (supplementary C.2 Fig. 6.(b)) and determined that after a certain number of iterations the tree structure remains stable.
>
> > 2. The choice of the smoothing factor $\beta$ in the reward update is not analyzed. Since directly affects how quickly MCTS adapts to new information versus retaining historical estimates, an ablation on its impact is necessary.
>
> We added the sensitivity analysis for $\beta$ in supplementary C.2 (Figure 5).
>
> > 3. Following the second point, the exploration parameter $\lambda$ is also not analyzed. And a corresponding analysis is necessary.
>
> In supplementary C.2 (Table 8 and Figure 6.a) we added sensitivity analysis for $\lambda$.
>
> > 4. The authors argue that the method performs conditional sampling; however, this is never explicitly materialized in the paper. In particular, the numeric form of $p(a_t|a<t)$ is not specified.
>
> We acknowledge that the specific numeric form of $p(a_t | a_{<t})$ was not explicitly written in a single equation. We clarify that an architecture is represented as a path through the learned hierarchy from the root to a leaf node. The joint probability of sampling an architecture is then factorized into the product of conditional probabilities following the sequence of nodes. Therefore, at depth $t$ of tree, the probability of sampling architecture $a_t$ given the previous path $a<t$, is determined by (eq. 2):
>
> $p(a_{i=t}|a_{<t}) = \frac{\exp(R(a_i)/ T)}{\sum_{j \in children(a_{t-1})} \exp(R(a_j)/ T)}$
>
> which is softmax probability all the children of the parent node $a_{t-1}$. From this distribution, we perform a categorical sampling step to select a specific branch $a_t$.
>
> > 5. The authors should analyze the impact of the chosen sampling strategy and evaluate alternative sampling methods under their framework.
>
> We thank the reviewer for their insightful suggestion. We chose UCT since it provides well-established theoretical guarantees.  The sampling strategy we apply during the warm-up is the standard uniform sampling strategy. In the main algorithm, we perform Boltzmann sampling (eq. 2) on standard UCT (eq. 3). This provides a degree of stochasticity compared to alternatively sampling with deterministic maximum reward. This helps prevent UCT from getting stuck in a sub-optimal branch.

---

### Author Response · Authors · 2026-02-15

We thank the reviewers for their time and valuable comments and questions. Their comments have contributed greatly to the improvement of our manuscripts. We are delighted that reviewers found our work addresses a relevant challenge for the TMLR audience.
We have revised the manuscript to reflect the questions and requested changes from the reviewers. The revised portions are shown in blue color in the manuscript.
The summary of the changes/additions are as follows:
* Addition of  “Generalizable and training-free NAS” to related work
* Notation change for eq. 4
* Addition of more methods for comparison to table 4
* Revision of “Limitations” to discuss specific bottlenecks
* Addition of “Broader Impact Statement”
* Addition of more details in appendix B.1 to specify the procedural choices such as batch size, distance metric and others
* Several additional ablations in appendix C.2 as requested by reviewers including:
  * Ablation on distance metric
  * Smoothing factor
  * Warm-up iterations
  * Exploration parameter
  * Clustering linkage
  * Validation batch
  * Temperature scheduling
* Proofreading and correcting typos and grammatical errors

---

> ### Comment · Action_Editor_K2oF · 2026-02-16
>
> Dear reviewers,
>
> Please look through the authors revision and see whether your concerns have been addressed. Thank you.
>
> Regards,
> Action Editor

---

### Comment · Action_Editor_K2oF · 2026-02-28
**Please look at the rebuttal by authors**

Dear reviewers,

Please look at the comments by the authors and provide more comments or your recommendation accordingly.

Regards,
AE

---

### Decision · Action_Editor_K2oF · 2026-03-13

**Recommendation:** Reject

**Audience:**

Yes

**Audience Explanation:**

The paper is on NAS, which is of clear interest to the ML community and TMLR readership.

**Claims And Evidence:**

No

**Claims Explanation:**

The paper lacks results on standard public NAS benchmarks (e.g., NAS‑Bench‑201/NATS‑Bench) to establish generality and fairness. Strong baselines are incompletely handled—key comparisons (e.g., MOTE‑NAS) show similar accuracy and far lower compute, yet no convincing justification is given for preferring the proposed method. The core “pre‑expansion” assumption and the claimed N<10k regime are only discussed in the text; no memory‑efficient implementation or experiments on larger search spaces are provided to demonstrate feasibility and scaling. The idea is promising but the evidence is insufficient for publication at TMLR.

**Resubmission Of Major Revision:**

The authors may consider submitting a major revision at a later time.